# Unveiling the Relationship between Ceftobiprole and High-Molecular-Mass (HMM) Penicillin-Binding Proteins (PBPs) in *Enterococcus faecalis*

**DOI:** 10.3390/antibiotics13010065

**Published:** 2024-01-09

**Authors:** Paola Conti, Lorenzo Mattia Lazzaro, Fabio Longo, Federica Lenzo, Alessandra Giardina, Sebastiano Alberto Fortuna, Stefania Stefani, Floriana Campanile

**Affiliations:** 1Department of Biomedical and Biotechnological Sciences (BIOMETEC), Section of Microbiology, University of Catania, 95123 Catania, Italy; paolacontias@hotmail.com (P.C.); lazzclml@gmail.com (L.M.L.); fabiolongo97@icloud.com (F.L.); fedelnz@outlook.it (F.L.); alessandragiardina27@gmail.com (A.G.); s.albertofortuna@gmail.com (S.A.F.); stefania.stefani@unict.it (S.S.); 2Department of Medical Biotechnologies, University of Siena, 53100 Siena, Italy; 3Department of Public Health and Pediatrics, University of Torino, 10126 Turin, Italy

**Keywords:** *Enterococcus faecalis*, ceftobiprole, PBPs, competition assays, Bocillin

## Abstract

Low-affinity PBP4, historically linked to penicillin resistance in *Enterococcus faecalis*, may still have affinity for novel cephalosporins. Ceftobiprole (BPR) is a common therapeutic choice, even with PBP4-related overexpression and amino acid substitution due to mutations. Our study aims to explore the interaction between BPR and High-Molecular-Mass (HMM) low-reactive PBPs in Penicillin-Resistant-Ampicillin-Susceptible/Ceftobiprole Non-Susceptible (PRAS/BPR-NS) *E. faecalis* clinical isolates. We conducted competition assays examining class A and B HMM PBPs from four PRAS/BPR-NS *E. faecalis* strains using purified membrane proteins and fluorescent penicillin (Bocillin FL), in treated and untreated conditions. Interaction strength was assessed calculating the 50% inhibitory concentration (IC_50_) values for ceftobiprole, by analyzing fluorescence intensity trends. Due to its low affinity, PBP4 did not display significant acylation among all strains. Moreover, both PBP1a and PBP1b showed a similar insensitivity trend. Conversely, other PBPs showed IC_50_ values ranging from 1/2-fold to 4-fold MICs. Upon higher BPR concentrations, increased percentages of PBP4 inhibition were observed in all strains. Our results support the hypothesis that PBP4 is necessary but not sufficient for BPR resistance, changing the paradigm for enterococcal cephalosporin resistance. We hypothesize that cooperation between class B PBP4 and at least one bifunctional class A PBP could be required to synthesize peptidoglycan and promote growth.

## 1. Introduction

As leading causes of multidrug-resistant hospital-acquired infections worldwide, enterococci represent the third most common cause of native valve endocarditis [1]. Among the *Enterococcus* genus, *Enterococcus faecalis*, and *Enterococcus faecium* are responsible for approximately 75% of all typed enterococcal infections [2]. Indeed, they are a common underlying cause of infections involving the urinary tract, abdomen, biliary tract, surgical wounds, as well as bacteremia, endocarditis, and burns. Their ability to thrive in healthcare settings is attributed to their intrinsic resistance to almost all cephalosporins, all semi-synthetic penicillins, aminoglycosides, clindamycin, and trimethoprim–sulfamethoxazole. In turn, this leads to several cases of multidrug resistance, extending hospitalization time, increasing the risk of treatment failure and death. Additionally, due to the remarkable genetic adaptability of enterococci, they can readily acquire new resistance traits, such as high-level aminoglycoside resistance, high-level ampicillin resistance, and vancomycin resistance, either through mutation or by horizontal gene transfer [3].

The production of β-lactamases, rare and unusual in enterococci, along with the expression of low-affinity Penicillin-Binding Proteins (PBPs), represent the two major mechanisms leading to their intrinsic resistance to most β-lactams [4]. PBPs are classified according to their masses as High-Molecular-Mass (HMM, >45 kDa) PBPs, comprising classes A and B, and Low-Molecular-Mass (LMM, <45 kDa) PBPs, consisting of class C PBPs only.

HMM PBPs are responsible for peptidoglycan polymerization and insertion into pre-existing cell walls [5]. Class A PBPs are bifunctional enzymes displaying glycosyltransferase and transpeptidase activity, whereas class B PBPs exert only transpeptidase activity [6]. The C-terminal binding domain (PB domain) of both classes catalyze peptide cross-linking between two adjacent glycan chains exerting transpeptidase activity. While the N-terminal domain of Class A PBPs displays glycosyltransferase activity only, catalyzing the elongations of uncross-linked glycan chains, in class B PBPs it is implicated in cell morphogenesis by interacting with other proteins involved in the cell cycle [7]. Class C PBPs possess peptidyl-transferases with D,D-carboxypeptidase and D,D-endopeptidase activity.

Enterococci harbor diverse PBPs, among which three belong to class A PBPs, and three to class B PBPs, while only two to class C PBPs [8]. Within the class B PBPs, low-affinity PBP4 *E. faecalis* and PBP5 *E. faecium* are known to play key roles in β-lactam resistance [9,10]. PBP4 has four structural domains: two N-terminal domains, known as the N1 domain, which anchors the enzyme to the cytoplasmic membrane, and an N2 domain, which may be involved in protein interactions; a C-terminal domain, with transpeptidase activity (TPase); and a non-penicillin-binding domain (nPB) [11]. Additionally, altered expression levels and mutations of genes encoding PBPs may also play a role in β-lactam resistance [12].

Ceftobiprole (BPR), a fifth-generation cephalosporin, has demonstrated a potent broad spectrum of activity against some Gram-positive bacteria, including *E. faecalis.* The high affinity of BPR to PBPs enables the drug to form a stable inhibitory complex, which determines its potent antibacterial activity.

The aim of this work was to analyze interactions occurring between BPR and High-Molecular-Mass (HMM) PBPs, with a particular focus on PBP4, to look for a possible role of PBP4 alterations influencing the binding to BPR using Penicillin-Resistant-Ampicillin-Susceptible/BPR Non-Susceptible (PRAS/BPR-NS) *E. faecalis* clinical isolates.

## 2. Results

### 2.1. Binding Profiles of HMM PBPs

Based on the PBP ATCC 47077 in silico analysis and the mobility on the gel, we detected six High-Molecular-Mass (HMM) and two Low-Molecular-Mass (LMM) PBPs (DD, carboxypeptidase). The exact molecular weight, the putative function of each protein, and the amino acid and nucleotide sequences listed in GenBank (https://www.ncbi.nlm.nih.gov/nuccore/CP025020.1 accessed on 13 February 2023) are shown in Table 1.

We further assessed the BPR 50% inhibitory concentration (IC_50_) values. Figure 1 and Table 2 report detailed BPR inhibition rates and PBP binding profiles of all *E. faecalis* strains in this study, along with corresponding Minimum Inhibitory Concentrations (MICs) values. Appendix A are also provided (Appendix A).

All strains showed similar behavior, whereby BPR consistently demonstrated low affinity for PBP4 in all isolates. Similar unresponsiveness was detected for PBP1b in almost all strains. Among the BPR non-susceptible strains (BPR-NS), E.fs7 (BPR MIC value 8 mg/L; range tested 4–32 mg/L), PBP1b and PBP4 were not inhibited by BPR, as demonstrated by their strong fluorescence to Bocillin FL, while BPR retained powerful affinity for PBP2, PBP2a, and PBP2b at different concentrations. Unexpectedly, E.fs7 lacked PBP1a. A similar inhibition profile was also recorded for E.fs8 (BPR MIC value 4 mg/L; range tested: 2–16 mg/L), but unlike E.fs7, an almost 50% inhibition was observed for PBP1b. PBP1a in E.fs18 (BPR MIC value 4 mg/L; range tested: 2–16 mg/L) did not undergo sufficient acylation even when their BPR inhibition rates were close to the IC_50_ value at 4× MIC; instead, PBP2, PBP2a, and PBP2b were readily inhibited at concentrations above 2× MIC. Despite the strong fluorescence shown in the gel, PBP1b and PBP4 were not significantly acylated at any concentration tested. In E.fs1 (BPR MIC value 16 mg/L; range tested 8–64 mg/L), PBP4 was inhibited by 4× MIC near to the IC_50_ value (45%). Conversely, other PBPs were adequately acylated, as reported by their IC_50_ values. (Appendix A).

Focusing on PBP4 binding to BPR, we observed that its inhibition percentages increased in all strains using a higher amount of the drug although never reaching IC_50_, reflecting that this inhibition is concentration-related and not MIC-concentration-related (related to the respective MIC value for each strain) (Figure 2).

### 2.2. Analysis of β-Lactamase Production

The hydrolytic activity of Nitrocefin vs. *S. aureus* ATCC 29213, used as Bla-positive control, was detected as an intense red color, demonstrating abundant enzyme release. On the other hand, *E. faecalis* ATCC 29212, as Bla-negative control, remains yellow, because of the absence of hydrolytic activity.

Efs1, Efs7, and Efs18 isolates developed a moderate red color 24 h after nitrocefin exposure, proving the enzyme production of β-lactamase and its activity. End-point PCR confirmed the presence of *bla*Z gene in these strains and were considered β-lactamase producers.

## 3. Discussion

Penicillin-resistant *E. faecalis* strains occur rarely but resistance can emerge following treatment with β-lactams [13]. Through the production of low-affinity PBPs, enterococci may acquire resistance to these antibiotics. Indeed, PBP4 in *E. faecalis* is considered a key player in intrinsic resistance against cephalosporins [8]. PBP4 operates peptidoglycan cross-linking, often in association with other PBPs such as PBP1a. Other PBPs such as PBP2a and PBP1b are thought to collaborate with PBP4, but direct interactions still require verification [14]. PBP2 is essential for cell viability. However, Dijoric et al. [15] demonstrated that PBP2, as well as PBP2b, display poor reactivity towards cephalosporins. Despite the low-affinity PBPs, BPR targets and binds these enzymes, with particular affinity for PBP4.

We related the in vitro antibacterial activity exerted by BPR to mutations of the *pbp*4 gene, and the affinity of BPR against PBPs. A complex mechanism of resistance to β-lactams emerged, whereby susceptibility varies for distinct β-lactams according to the PBP class with which PBP4 collaborates.

A link between resistance to benzyl-penicillins and non-susceptibility to fifth-generation cephalosporins such as BPR emerged, related to deletion of an adenine in the regulatory region upstream of the *pbp*4 promoter, causing altered expression levels of the *pbp*4 gene, along with missense mutations, destabilizing the PBP/β-lactam complex. Previous reports have described mutations in the *pbp*4 gene [16,17,18]. Indeed, Rice et al. [13] and Lazzaro LM et al. [12] observed that adenine deletion (*del*A) −35 bp upstream of the promoter region led to an over-expression of the *pbp*4 gene due to an alteration of the binding of regulatory proteins. As a result, increased levels of transpeptidation in peptidoglycan leading to higher levels of cross-linking in the bacterial cell wall are reached, resulting in higher MIC values. Consistently with these findings, the *del*A mutation carried by the four BPR-NS strains could explain the in vitro non-susceptibility, albeit still retaining bactericidal activity (Table 3) [19]. Missense mutations near the serine active site and the catalytic motifs of the enzyme may also play a role in increasing the MIC value. Such substitutions could destabilize the BPR/PBP complex, leading to a less efficient one. This was particularly true in PRAS/BPR-NS E.fs1, whereby the T418A mutation falls six amino acids upstream of the catalytic serine of motif I, leading to a BPR MIC value of 16 mg/L, and bactericidal activity only at 4× MIC. These results also led to the hypothesis that, due to the low affinity of PBP4, initial binding to BPR does not occur (explaining the in vitro antibacterial activity) but can be seen upon prolonged exposure and at higher concentrations of BPR.

PBP4 expression induction by BPR was not assessed in this study. Further work should also elucidate whether BPR impacts PBP4 and other PBP expression profile in different phenotypic backgrounds carrying advantageous gene mutations.

Regarding the PBP binding profiles and BPR inhibition rates analyzed by IC_50_ values, the major role of PBP4 as necessary and sufficient PBP conferring high-level resistance to β-lactams was not confirmed. Indeed, PBP4 is the only PBP lacking acylation, but this trend is similar to that observed for PBP1b (only acylated in E.fs1, at BPR concentration above the MIC value). Thus, it is tempting to speculate that PBP4 requires the cooperation of a bifunctional PBP to determine cephalosporin resistance.

The labeling of PBPs with Bocillin FL in competition assays is dependent on the availability of PBPs for covalent interaction with the fluorescent penicillin. The overexpression of low-affinity PBP4 could contribute to its inability to bind effectively with β-lactams and, even after treatment with increasing ceftobiprole concentrations, there may always be enough PBP4 amount available for covalent interaction with Bocillin. Moreover, the labeling efficiency of PBP4 with Bocillin can vary among the strains, according to their different levels of expression.

Taken together, no direct correlation between BPR MIC values, expression levels of *pbp*4, aminoacidic substitutions in the catalytic site [12], and IC_50_ values were found in this subset of *E. faecalis* strains. BPR binding affinity to PBP4 is concentration-dependent and does not influence MIC values. Moreover, BPR-NS strains showed a higher amount of PBPs other than PBP4 and PBP1b inhibited by BPR. Over-expressed PBP4 and PBP1b may be sufficient to cause reduced susceptibility to BPR. The impact of other PBP mutations should also be further analyzed to clarify their relationship in BPR resistance in *E faecalis*.

Moreover, although rare [20,21], the three strains were found to be weak β-lactamase producers. While the acyl–enzyme complex is initially stable, the production of β-lactamases in E.fs1, E.fs7, and E.fs18 could catalyze the deacylation reaction, leading to the inactivation and degradation of the antibiotic molecule, thus enhancing resistance against β-lactams and potentially also against fifth-generation cephalosporins. Ceftobiprole should be more prone to degradation by β-lactamase activity in E.fs1, E.fs7, and E.fs18, reducing their inhibitory potency. In contrast, Bocillin FL, being a modified penicillin, may have enhanced stability against such degradation mechanisms, leading to lower MIC values.

## 4. Materials and Methods

### 4.1. Selected Strains

Four *E. faecalis* strains were selected for their antibiotic-resistance behaviors, belonging to the major multi-drug resistance phenotypes: Penicillin-Resistant Ampicillin-Susceptible (PRAS), Ceftobiprole Non-Susceptible (BPR-NS), Vancomycin-Resistant (VRE), and High-level Aminoglycoside Resistant of (HLAR), from a previously characterized collection of seven *E. faecalis* clinical strains, isolated from bloodstream infections (BSI) in Italian hospitals, showing diverse mutations in coding and non-coding regions of *pbp*4 [12].

The MIC values were determined by the reference broth microdilution method and interpreted according to the European Committee on Antimicrobial Susceptibility Testing (EUCAST) clinical breakpoints (http://www.eucast.org/clinical_breakpoints/ accessed on 1 January 2021) (EUCAST, 2021). In the absence of EUCAST clinical breakpoints, those of the Clinical and Laboratory Standards Institute were applied (Clinical and Laboratory Standards Institute, 2021) (Table 3).

Two β-lactam-susceptible *E. faecalis* strains were also included as control: *E. faecalis* ATCC 29212, used as control for antimicrobial susceptibility testing [22], and *E. faecalis* OG1-RF, deposited in the American Type Culture Collection (ATCC) under ATCC 47077, deriving from *E. faecalis* OG1 by selection for resistance to rifampin and fusidic acid [23], used as control in the competition assay study.

### 4.2. Study of PBP4/BPR Affinity and Competition Assay

To evaluate the affinity of BPR for PBPs and determine if PBP4 significant aminoacidic substitutions could affect and alter the binding and stability of the BPR/PBP4 complex, competition assays were carried out. We used Bocillin FL (Invitrogen-life Technologies, Turin, Italy), a fluorescent β-lactam composed of penicillin V derivative linked to BODIPY fluorochrome. Due to its composition, Bocillin FL reacts with empty binding sites that remain after the reaction of PBPs with unlabeled BPR, detecting the formation of the β-lactam/PBP complex.

Therefore, through the evaluation of the Bocillin fluorescence level at different concentrations of BPR tested, we assessed the antibiotic concentration that displaces 50% of the labeled molecule from the binding site (IC_50_). Thus, we obtained the potential rate of inhibition of PBP acylation in PRAS/BPR-NS strains.

The relative binding of BPR was evaluated at four different concentrations (1/2-fold to 4-fold the MIC value, specifically 1/2×, 1×, 2×, and 4 × MIC).

Briefly, a bacterial colony from fresh culture plate was inoculated into 5 mL of BHI broth and incubated o/n at 37 °C. The strain culture was measured to an OD_450_ nm to obtain a starting inoculum of 10^5^–10^6^ CFU/mL. The cells were therefore incubated for two hours at 37 °C, until reaching an OD_620_ = 0.20, corresponding to the exponential phase. Bacteria were harvested by centrifugation for 2 min at room temperature (RT) and washed in PBS (10 mM, pH 7.4). Different BPR concentrations were added. The samples were incubated at 37 °C for 30 min and resuspended in 50 μL of PBS containing 5 μg/mL of Bocillin FL (7.6 μM).

Following incubation with lysozyme (10 mg/mL) for 30 min at 37 °C and 5 cycles of sonication in a Banderlin Sonifier (30 s, 30% duty cycle for 6 intervals), the protein concentration was determined by using the Qubit^®^ protein Assay Kit, a fluorometric test executed using Qubit 2.0 (Invitrogen-life Technologies, Turin, Italy), and normalized at the same protein concentration; a 51 μL measure of each sample was added with 17 μL of 4× Laemmli buffer *plus* β-mercaptoethanol, as a reducing reagent, boiled for 5 min at 90 °C for protein denaturation, and cooled at RT. Electrophoresis was carried out in an SDS–PAGE on a Criterion™ TGX™ (Tris-Glycine extended) 8–16% polyacrylamide gel, for 2 h at a constant voltage of 100 V. To predict the molecular weight of each PBP, two different commercial pre-stained standards, SeeBlue^®^ Plus2 Prestained Standard (Invitrogen by Thermo Fisher Scientific, Basingstoke, UK) and Thermo Scientific Spectra Multicolor High Range Protein Ladder, were used. The fluorescent Bocillin FL covalently bound to the PBPs was detected with excitation at 488 nm and emission at 520 nm (Typhoon FLA 9500; filter Alexa 488; PMT 1000; pixel 50 μm, Washington, DC, USA). To detect the PBP profile and identify their molecular weight, the SDS–PAGE Criterion™ TGX™ (Hercules, CA, USA) (Tris-Glycine extended) 8–16% polyacrylamide gel was stained for 1 h in a Coomassie brilliant blue R-250 buffer (40% methanol; 10% glacial acetic acid; 0.1% Coomassie Brilliant Blue R-250, St. Louis, MO, USA), and de-stained for 1 h (40% methanol; 10% glacial acetic acid).

The fluorescence was analyzed using ImageJ software (https://imagej.net/ij/features.html, accessed on 25 November 2023). The brightness and contrast on the image should be adjusted to optimize the signal-to-noise ratio. The background signal should be subtracted. Normalization was obtained correlating labeled PBPs with Coomassie-stained PBPs.

Our study was completed by the determination of the relative binding (IC_50_), i.e., the 50% inhibitory concentration values (concentration of ceftobiprole (μg/mL) needed to reduce by 50% the binding of Bocillin FL to individual PBPs), determined by plotting the PBP band volumes versus compound concentrations. The 100% binding of Bocillin FL was represented by the PBPs labeled with Bocillin FL but with no drug. IC_50_ values were calculated and statistically analyzed using the GraphPad Prism 8 software. We performed an in silico analysis for ATCC 47077 PBPs. We calculated the exact molecular weight of each PBP by the online bioinformatic database “Quest Calculate™ Peptide and Protein Molecular Weight Calculator” (AAT Bioquest, Inc.) (https://www.aatbio.com/tools/calculate-peptide-and-protein-molecular-weight-mw, accessed on 25 November 2023) and the amino acid and nucleotide sequences listed in GenBank (https://www.ncbi.nlm.nih.gov/nuccore/CP025020.1, accessed on 25 November 2023). The putative function of each protein was also inferred by the NCTC database.

### 4.3. β-Lactamase Assay

β-lactamase activity was evaluated by nitrocefin assay (OxoidTM, Thermo ScientificTM, Basingstoke, UK). Even though nitrocefin is a cephalosporin, and thus possesses a β-lactam ring susceptible to β-lactamase mediate hydrolysis, it does not have antimicrobial properties. Furthermore, if degraded, nitrocefin rapidly changes color from yellow to red.

Starting from a high inoculum (10^9^ CFU/mL), simulating in vivo infections, cells were incubated for 24 h with and without oxacillin at a final concentration of 8 mg/mL. The detection of the β-lactamase activity on the supernatant and pellets (cellular debris) was performed using already published protocols [21], repeated in duplicate, and reconfirmed. *S. aureus* ATCC 29213 was used as positive control; *E. faecalis* ATCC 29212 was used as Bla-negative control. End-point PCR for *bla*Z gene was executed designing primers on the sequence of *S. aureus* ATCC 29213, the Bla+ control strain.

## 5. Conclusions

In conclusion, in PRAS/BPR-NS strains, mutations of the *pbp*4 gene (*del*A and amino acid substitutions) may influence the susceptibility profile to β-lactams, and particularly to BPR, increasing the MIC value and influencing the binding to the antimicrobial, without altering the in vivo bactericidal activity. Furthermore, the low affinity of PBP4 is a common feature of all strains, regardless of catalytic site alterations. In addition, other PBPs may be involved in intrinsic cephalosporin resistance in *E. faecalis*. PBP4 was necessary but not sufficient to determine BPR resistance, and β-lactam exposure requires cooperation of PBP4 and, at least, another class A PBP (preferably PBP1b). Moreover, the production of β-lactamases may contribute to enhance resistance to β-lactams and potentially also to fifth-generation cephalosporins.

Further analyses are needed to evaluate how PBPs other than PBP4 are involved in non-susceptibility to BPR, and what mutations and/or altered expression levels are required. Regarding PBP4, the role of *del*A and aminoacidic substitutions via site-direct mutagenesis should be analyzed, along with in vivo expression levels of *pbp*4 and BPR/PBP affinity at different exposure times and concentrations.

## Figures and Tables

**Figure 1 antibiotics-13-00065-f001:**
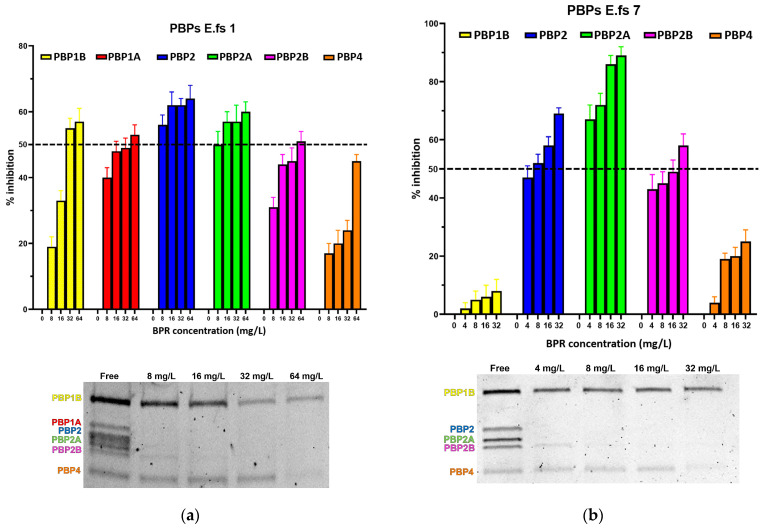
Relative affinities of BPR (above) and PBP/Bocillin-labeled profiles on SDS–PAGE gel (below) for BPR-NS *E. faecalis* clinical isolates (**a**) E.fs1, (**b**) E.fs7, (**c**) E.fs8, and (**d**) E.fs18. Error bars indicate the average of three biological replicates in three assays ± SD.

**Figure 2 antibiotics-13-00065-f002:**
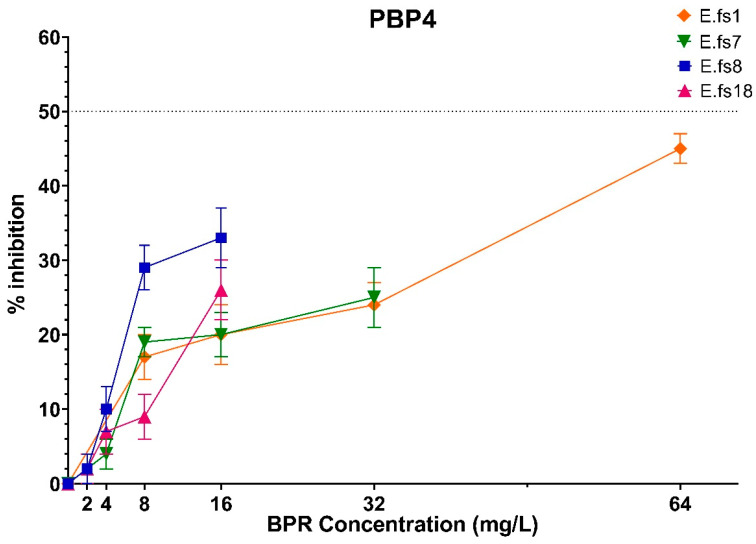
Comparison of PBP4 percentages of inhibition concentrations of BPR of all strains. Orange line: E.fs1, Green line: E.fs7, Blue line: E.fs8, Fuchsia line: Efs18. GraphPad Prism 8 software (version 8.4.0).

**Table 1 antibiotics-13-00065-t001:** PBP genes of *E. faecalis* OG1RF (ATCC 47077) chromosome, corresponding proteins, and putative molecular weights (kDa) (this study).

PBP	PBP (Other Name)	PBP Class	^a^ Locus_Tag	N. Amino Acids (aa)	^b^ Molecular Weight (kDa)	^c^ Enzymatic Activity	Gene	GenBank CP025020.1 (Position)	N. Nucleotides (bp)
PBP1B	PBPZ	class A	CVT43_07655	803	88.40413	TPase/Gtase	*pbp1b*	1509774-1512185	2412
PBP1A	PONA	class A	CVT43_04970	778	85.35237	TPase/Gtase	*pbp1a*	431566-433752	2331
PBP2	PBPB	class B	CVT43_03860	742	81.71274	TPase—Cell division protein FtsI	*pbp2*	747766-749994	2229
PBP2A	PBPF	class A	CVT43_02290	728	79.56664	TPase/Gtase	*pbp2a*	966441-968777	2187
PBP2B	PBPA	class B	CVT43_11360	711	77.85899	TPase	*pbp2b*	2276447-2278582	2136
PBP4	PBP4(5)	class B	CVT43_10030	680	74.01754	TPase	*pbp4*	2010904-2012946	2043
PBP	PBP	class C	CVT43_05000	498	57.71227	D,D Carboxypeptidase M32	*pbp M32*	972441-973937	1497
VanY	PBP	class C	CVT43_11375	236	27.04	D-Ala-D-Ala carboxypeptidase VanY	*pbp vanY*	2281379-2282089	711

^a^ https://www.ncbi.nlm.nih.gov/nuccore/CP025020.1, accessed on 25 November 2023. ^b^ PBP molecular weights were calculated with the online bioinformatic software AAT Bioquest, Inc. 13 February 2023, Pleasanton, CA, USA (1 February 2023). Quest Calculate™ Peptide and Protein Molecular Weight Calculator. AAT Bioquest. ^c^ Transpeptidase (TPase); Transpeptidase/Glycosyltransferase (TPase/Gtase).

**Table 2 antibiotics-13-00065-t002:** Inhibition of PBPs from the *E. faecalis* strains in this study, and determination of IC_50_ value.

Strain	IC_50_ (μg/mL) PBP						MIC BPR (mg/L)
	1B	1A	2	2A	2B	4	
E.fs1	28.59	38.45	7.30	8.42	57.64	N.A.	16
E.fs7	N.A.	N.A.	7	3.07	16.3	N.A.	8
E.fs8	N.A.	1.53	1.53	0.31	2.2	N.A.	4
E.fs18	N.A.	N.A.	14.2	15.19	14.2	N.A.	4

IC_50_, Concentration of the β-lactam antibiotic that inhibits 50% of the Bocillin FL, in comparison with the level for the control containing no drug. N.A.: not applicable.

**Table 3 antibiotics-13-00065-t003:** Beta-lactam and comparator drug MIC values (mg/L) and PBP4 sequence alterations of the four *E. faecalis* strains included in this study.

Code	MDR-Phenotype	MIC Values (mg/L)								Deletion in Promoter Region ^b^	Amino Acid Substitutions in PBP4 ^d^											
													PBP Active Sites									
		P ^1^	AMP	AML	IMI	BPR ^2^	CPT ^2^	VA	TEC		_50_T	_223_I	_418_T	_475_L	_536_A	_573_D	_605_Y	_606_V	_639_L	_665_T	_666_D	_678_T
**E.fs1**	**PRAS-BPR-NS-HLAR**	16	1	0.5	4	16	>256	0.5	2	2013028_2013029 *del*A ^c^	I	-	A	-	T	-	-	-	-	-	P	-
**E.fs7**	**PRAS-BPR-NS-HLAR**	64	4	4	4	8	>256	1	2	2013028_2013029 *del*A ^c^	I	-	-	-	-	-	-	A	F	I	-	A
**E.fs8**	**PRAS-BPR-NS-VRE ^a^**	16	4	1	2	4	32	>256	>256	2013028_2013029 *del*A ^c^	I	-	-	Q	-	-	-	-	-	-	-	-
**E.fs18**	**PRAS-BPR-NS**	16	2	1	2	4	4	1	2	2013028_2013029 *del*A ^c^	-	-	-	-	-	-	H	-	-	-	-	-

^1^ Penicillin susceptibility values were established according to CLSI breakpoints (EUCAST breakpoints absent). ^2^ Ceftobiprole and Ceftaroline: No EUCAST and CLSI official breakpoints; eCOFFs not determined. P: Penicillin; AMP: Ampicillin; AML: Amoxicillin; IMI: Imipenem; BPR: Ceftobiprole; CPT: Ceftaroline; VA: Vancomycin; TEC: Teicoplanin. ^a^
*van*A; ^b^ a single base pair deletion 8 bases upstream of the putative −35 region; ^c^ Accession number GenBank: CP025020.1 (ATCC47077); ^d^ Protein ID GenBank: AEA94594.1 (ATCC47077). PRAS: Penicillin-Resistant Ampicillin-Susceptible; BPR-NS: Ceftobiprole Non-Susceptible; HLAR: High-Level Aminoglycoside Resistance; VRE: Vancomycin-Resistant *E. faecalis*. PBP4 sequences of the strains in this study were submitted in GenBank with ID accession no. OM032878 (Efs7); OM032880 (Efs1); OM032881 (Efs8); and OM032883 (Efs18) [12].

## Data Availability

PBP4 sequences of the strains in this study were submitted to GenBank with ID accession no. OM032878 (Efs7); OM032880 (Efs1); OM032881 (Efs8); and OM032883 (Efs18).

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
