# Peer review of "Unveiling the Relationship between Ceftobiprole and High-Molecular-Mass (HMM) Penicillin-Binding Proteins (PBPs) in Enterococcus faecalis"

_antibiotics, 2024, doi:10.3390/antibiotics13010065_

Round 1
Reviewer 1 Report
Comments and Suggestions for Authors
Paola et al. described the relationship between ceftobiprole and PBPs in Enterococcus faecalis. After competition assays, the authors concluded that PBP4 is necessary but not sufficient for BPR resistance, changing the paradigm for enterococcal cephalosporin resistance. This work is interesting and fits the scope of Antibiotics. However, the research is quite superficial. Several factors can induce resistance, involving PBP overexpression, heterogeneous low-affinity PBP acquisition, PBP binding site mutation, etc. In this manuscript, the authors only examined the binding affinity between PBP4 and ceftobiprole. Some issues still need to be explained. I suggest the manuscript can be published after a major revision.
1. Does ceftobiprole affect PBP4 expression in E. faecalis? The related data should be determined.
2. Ceftobiprole can compete with bocillin FL to bind to PBP4. What is the binding site of Ceftobiprole in PBP4? The molecular docking might help to find the possible mutation amino acid that induces affinity change.
3. Proteomics can provide insight view of evaluating whether PBP4 or other PBPs are vital for Ceftobiprole resistance. The results can also confirm the authors' conclusion.
4. Table 2, The MIC of ceftobiprole against E.fs18 is 4 mg/L, but the IC50 against PBP is far above 4 mg/L (esp. 14.2,15.19, 14.2mg/L to PBP2, 2A, 2B). Are there other antibacterial mechanisms of ceftobiprole except for binding to PBP and inhibiting peptidoglycan synthesis? Similar result exists in E.fs1, E.fs7.
Reviewer 2 Report
Comments and Suggestions for Authors
This manuscript, authored by Paola and colleagues, discusses an interesting approach to study the interactions between the antibiotic ceftobiprole (BPR) and the penicillin-binding proteins (PBPs) in the cell walls of Enterococcus faecalis. The authors specifically study the interactions of BPR with different PBPs using competitive drug binding experiments and electrophoretic analyses. The key observation of the study was that BPR has low affinity for the PBPs – PBP1b and PBP4. It was concluded that the higher fluorescence observed for these proteins in SDS PAGE was due to their acylation by Bocillin, which competed with BPR for the binding sites; thus, indicating the low affinity of BPR for PBP4 and PBP1b respectively.
The design of the study is simple and has the potential to produce valuable data. Particularly, I appreciate the thought of studying different PBPs using fluorophores and a simple experiment such as SDS-PAGE. However, I believe that there are minor flaws in the choice of reagents used, and the interpretation of the obtained data. The reasons for these opinions are listed below. It is for these reasons that I recommend publishing the article with major changes.
Major corrections:
1. Authors discuss extensively that Bocillin competes with BPR for binding with PBPs. However, BPR does not bind non-covalently. Since this interaction leads to the production of a stable ester bond, Bocillin cannot ‘displace’ BPR from PBPs. In the discussion and conclusion sections, authors propose acylation-deacylation cycles for BPR – if this is true, Bocillin still will not ‘complete’ for the sites as the deacylated protein will not regenerate BPR (BPR remains hydrolyzed and is permanently lost). For this reason, it is my opinion that Bocillin only reacts with unreacted sites, left out after the reaction of PBPs with BPR. Is there any literature on diacylation of PBPs – if yes, authors may cite such papers in the manuscript?
2. L155-L167: Authors discuss that overexpression of PBP4 is frequently observed in resistant strains. This could be the reason why PBP4 is always labelled by Bocillin in competitive assays. The bacteria might have contained very high amounts of PBP4 to begin with, and the treatment with BPR might not have consumed the PBP4 pool. Thus, there is always PBP4 available for covalent reaction with Bocillin. Also, it could be for this reason that the authors did not observe an MIC for PBP4, as mentioned at L195-L196.
3. L243-L244: Authors mention that Bocillin displaces BPR from its binding site. If the authors cannot find a reference for this deacylation process, the sentence may have to be edited. As discussed earlier, this is a covalent reaction and most likely, it cannot be displaced.
Since beta lactams like BPR are only bacteriostatic, they do not inhibit protein synthesis. Therefore, BPR treatment may induce the production of new PBPs in the bacteria as a survival mechanism (to compensate for the loss of PBPs). Therefore, Bocillin may label the newly produced PBPs and actually not displace BPR from the modified PBPs.
4. MICs have been traditionally determined by counting viable cells (after antibiotic treatment) by either counting the colonies on an agar plate, or spectrophotometric methods using dyes such as resazurin. The authors determined the MIC of BPR for these isolates using competitive assay – which actually may not be competitive as discussed in the previous comments. So, are the MICs determined by this method comparable to the values reported in literature, determined by any of the traditional methods? It may be useful to cite these papers here.
Minor corrections:
1. Deacylation versus diacylation (spell check) in discussion and conclusion sections. Deacylation is the removal of acyl moiety. Diacylation is acylation done twice.
2. There is enough structural difference between Bocillin and BPR. Readers will be convinced to see literature cited, which discusses the equality or differences between the Kds of these compounds?
Round 2
Reviewer 1 Report
Comments and Suggestions for Authors
The manuscript can be accepted in the present form.
Reviewer 2 Report
Comments and Suggestions for Authors
I thank the authors for considering my comments on the manuscript. In my opinion, the manuscript is now complete and ready for publication.